# Public Sentiment on Governmental COVID-19 Measures in Dutch Social Media

**Shihan Wang[1], Marijn Schraagen[1], Erik Tjong Kim Sang[2], and Mehdi Dastani[1]**

[1]Department of Information and Computing Sciences, Utrecht University
[2]Netherlands eScience Center
The Netherlands
s.wang2@uu.nl, m.p.schraagen@uu.nl,
e.tjongkimsang@esciencecenter.nl, m.m.dastani@uu.nl

## Abstract

Public sentiment (the opinion, attitude or feeling that the public expresses) is a factor of interest for government, as it directly influences the implementation of policies. Given the unprecedented nature of the COVID-19 crisis, having an up-to-date representation of public sentiment on governmental measures and announcements is crucial. In this paper, we analyse Dutch public sentiment on governmental COVID-19 measures from text data collected across three online media sources (Twitter, Reddit and Nu.nl) from February to September 2020. We apply sentiment analysis methods to analyse polarity over time, as well as to identify stance towards two specific pandemic policies regarding social distancing and wearing face masks. The presented preliminary results provide valuable insights into the narratives shown in vast social media text data, which help understand the influence of COVID-19 measures on the general public.

## 1 Introduction

Public sentiment (the opinion, attitude or feeling that the public expresses) can directly influence the implementation of policies (Burstein, 2003), therefore it is crucial for policy makers to know the public sentiment of chosen policies and to take this sentiment into account when deciding on new policies. Given the unprecedented nature of the COVID-19 crisis, having an up-to-date representation of public sentiment on governmental measures and announcements becomes even more important. However, the 'staying-at-home' policy makes analysing public sentiment by means of face-to-face research methods like interviews and questionnaires challenging, while classical online surveys could delay the analysis results by restrictions of frequency.

With the rapid growth of online social media, monitoring public sentiment on platforms like Twitter and Reddit allows for much more and frequent measurements and a better indication of changes over time (Tan et al., 2013; Wang and Terano, 2015). Thus, we apply natural language processing (NLP) approaches on Dutch social media to understand the temporal variation of Dutch public sentiment during the COVID-19 outbreak period.

Our analysis covers two perspectives of sentiment analysis: polarity analysis (whether a message is positive or negative) and stance analysis (whether a message is supportive of or against a given target (Li and Caragea, 2019)). In our study, the given targets are policy measures taken by the Dutch government. We particularly focus on the public attitude towards two specific measures, i.e., social distancing and wearing face masks. To validate our work for the broader Dutch public, we collected data from three different online media platforms (Twitter, Reddit and Nu.nl) to perform a comparison study. The preliminary results of analysis are presented in this paper. As a summary of our contributions, we provide a first sentiment-oriented overview of Dutch public discussion around COVID-19 across multiple social media sources and explore the practical usage of NLP approaches for understanding the influence of COVID-19 measures on the general public.

## 2 Related work

Social media sentiment has previously been analyzed during pandemics such as H1N1(Chew and Eysenback, 2010). Also the COVID-19 pandemic, despite being a relatively new topic, has attracted many researchers from different areas, including social media analysis. Abd-Alrazaq et al. (2020) identified four main COVID-19 related themes on Twitter: virus origin, contamination sources, preventive measures and impact on societies. Zhao et al. (2020) identified topics and sentiment related to COVID-19 in China. Samuel et al. (2020) con-

ducted textual analyses of Twitter COVID-19 data to identify public fear sentiment. Several studies focused on bots (Ferrara, 2020) and misinformation (Kouzy et al., 2020; Singh et al., 2020) which influence opinions on social media. Chen et al. (2020) showed that positive polarity government messages on social media result in higher public engagement. Cinelli et al. (2020) identified COVID-19-related topics in various social media, including Reddit and Twitter. They found that the ratio of misinformation to reliable information was found to be stable over time, however Twitter has a larger percentage of misinformation compared to Reddit.

From a technical point of view, stance detection has recently received considerable attention in the NLP community (Küçük and Can, 2020), for which neural networks with word embeddings have proven to be effective (Yi-Chin Chen, 2017; Li and Caragea, 2019). An annotation and training approach for classifying hate speech in COVID-19 related tweets using embeddings was described by (Cotik et al., 2020). Alternative approaches include unsupervised clustering for stance analysis (Darwish et al., 2020).

## 3 Methodology

### 3.1 Data collection

Data for polarity and stance analysis is collected from Twitter, Reddit and Nu.nl. For Twitter, all Dutch language tweets are collected via the Twitter streaming API (using the provided *lang* attribute) and are subsequently filtered using a set of keywords related to COVID-19 (as presented in Table 1). Reddit and Nu.nl are organized by topic, so for these two data sources all messages from Corona-related threads are used without keyword filtering. Nu.nl is a news website that allows people to comment on news articles and blogs. From this data source, our analysis concentrates on the comments instead of the contents of articles. The timespan of the datasets is February 27th 2020 (when the first COVID-19 patient was discovered in the Netherlands) until September 2020. The amount of collected messages is presented in Table 2.

### 3.2 Data annotation and analysis

The goal of the analysis is to provide both general polarity analysis for COVID-19 related messages, and stance analysis towards particular subtopics. For polarity analysis we use the library `pattern.nl` (Smedt and Daelemans, 2012), for

| Category | Keyword | Translation |
|----------|---------|-------------|
| Disease | corona | |
| | covid | |
| Health care | huisarts | doctor |
| | mondkapje | face mask |
| Government | rivm | national health organization |
| Social | flattenthecurve | |
| | blijfthuis | stay home |
| | houvol | hang in there |

Table 1: COVID-19 keywords for filtering topic tweets.

| Month | Tweets | Nu.nl | Reddit |
|-------|--------|-------|--------|
| February | 278,082 | 25,721 | 5 |
| March | 3,152,638 | 207,957 | 28,038 |
| April | 2,115,728 | 193,530 | 11,943 |
| May | 1,264,650 | 146,832 | 6,452 |
| June | 921,481 | 90,698 | 3,218 |
| July | 922,992 | 85,085 | 2,985 |
| August | 1,078,644 | 105,047 | 4,738 |
| September | 1,115,057 | 114,480 | 6,486 |

Table 2: Number of messages per dataset over time.

which no training is required. This library contains a lexicon of 3918 Dutch polarity words, mostly adjectives, and 120 language-independent emoji.

For stance analysis, we trained classifiers based on manually annotated data. Two topics related to governmental measures are selected, the social distancing measure (i.e. all people keep 1.5 metres distance from each other except when they are living in the same house) and whether or not the government should enforce face mask use by the general public (the Dutch government opposed face mask wearing until recently). We define three possible labels (*support, reject, other*) for both topics.

For the social distancing measure messages are selected using the following pattern: *anderhalve* (one and a half) followed by *meter*, or *1.5* or *1,5* followed by *m*, or *afstand* (distance) and *hou* (keep) anywhere in the message in any order.[1] This resulted in 994,052 tweets, 2,930 Reddit comments and 40,429 Nu.nl comments. In total, 5,732 messages (randomly selected from tweets data) were manually annotated by a single annotator (a second annotator was involved to validate the annotation results), answering the question *Does the message*

---

[1]The following regular expression was implemented:
```
1[.,]5[ -]*m|afstand.*hou|hou.*afstand
|anderhalve[ -]*meter
```

*support or reject the social distancing measure announced by the Dutch government on 15 March 2020?*

For the face mask discussion, messages containing the word *mondkapje* (face mask) were selected. 578 tweets and 744 Nu.nl comments have been annotated manually by the same annotator, answering the question *Does this message support or reject the policy on advising against the use of face masks by the general public?* Note that a rejection of the face mask policy actually entails that the person is in favor of (mandatory) face masks.

Using the annotated data, for each topic a classifier is trained with the fastText library for Python (Bojanowski et al., 2017; Joulin et al., 2017). The fastText classifier is a linear feed forward network trained using stochastic gradient descent. The network contains an embedding layer for subword features. In our current experiments the subword embeddings are trained on the fly, as preliminary experiments showed that using pre-trained embeddings did not increase the performance. After training and evaluating these models, the classifiers are used to predict the stance of all other messages in the dataset in order to perform a comprehensive stance analysis for the target topics.

## 4 Results

### 4.1 Polarity analysis

We computed daily average polarity score for three data sources and found that trends of public polarity have stable fluctuations over time (for instance the results of Twitter data are shown in Figure 1). Some interesting links can be found between press conferences and trends of public sentiment. For instance, general polarity scores reached a minimum at the March 12th press conference of the Dutch government (A in Figure 1), when the first lockdown measures were announced. More recently, the COVID-19 related polarity reached a peak on May 19th (B in Figure 1), when the government announced the first release measures. In addition, the COVID-19 related tweets is generally more negative than general Dutch tweets, while the polarity score regarding social distancing was decreasing in the first months. Similar results were found in the other two data sources.

### 4.2 Stance analysis training

For training the fastText model for stance analysis, we used ten-fold cross-validation on the human-

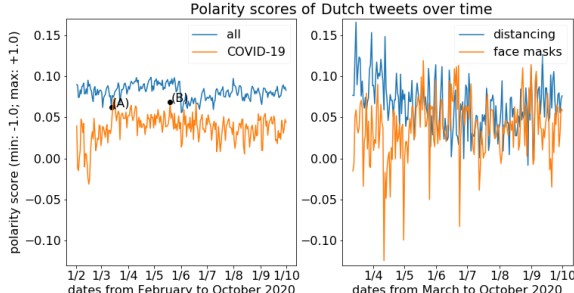

Figure 1: Daily average polarity scores of all Dutch tweets, the COVID-19 topic related tweets and two measures-related tweets.

labeled tweets in combination with a grid search to determine the optimal word vector dimension (10-300), number of training epochs (10-500) and learning rate (0.05-1.0). For the setup of the neural network itself we used the default settings of the fastText library. The dataset was divided in 80% training, 10% validation (for parameter optimization), and 10% test examples. Table 3 shows the parameter settings and classification accuracy results. The baseline method labels the given message as the majority class (*support* for distancing and *reject* for face masks).

|  | distancing | face masks |
|---|---|---|
| vector length | 10 | 300 |
| learning rate | 0.2 | 0.2 |
| epochs | 10 | 10 |
| baseline accuracy | 0.56 | 0.42 |
| validation accuracy | 0.65 | 0.56 |
| test accuracy | **0.65** | **0.55** |

Table 3: Stance classification parameters and results

An additional experiment was performed to investigate the effect of training set size on classification accuracy. As shown in Table 4, this experiment showed that the classifier was almost linearly improving with training size, therefore adding more annotated data is expected to increase the accuracy further.

| training set size | distancing | face masks |
|---|---|---|
| 100 | 0.56 | 0.46 |
| 200 | 0.56 | 0.49 |
| 500 | 0.58 | 0.52 |
| 1,000 | 0.60 | 0.55 |
| 5,000 | 0.65 | - |

Table 4: The relation between training set size and classification performance, measured by accuracy

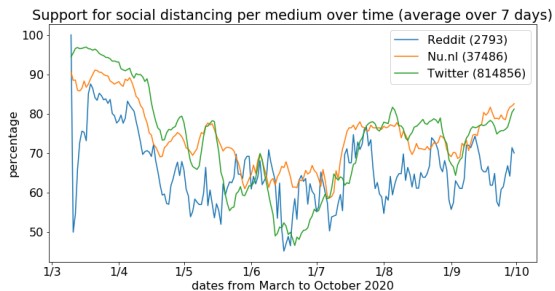

Figure 2: Development of public support for the March policy on social distancing.

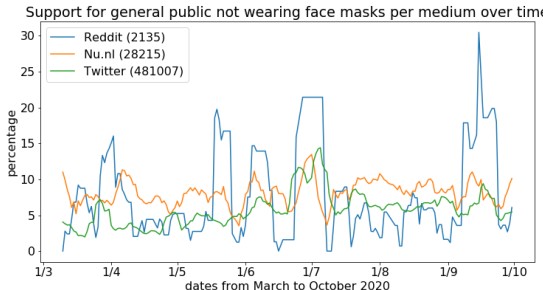

Figure 3: Development of public support for the March policy on face mask wearing.

## 4.3 Stance analysis application

We applied the trained classifier for the social distancing topic to all data and present the results in Figure 2. A similar trend among three platforms can be observed that validates our stance analysis results (due to the small size of Reddit data, its results may contain considerable noise). The public support is initially high in March, however it gradually decreases in the following months (until June 15th). This is consistent with reports of the Dutch health authority RIVM (RIVM, 2020a). Figure 2 also shows an increasing support for the social distancing measure after June 15th, which has been confirmed by national questionnaire results in September (RIVM, 2020b). This finding supports the validity and timeliness of our analysis.

The classifier trained on annotated face mask-related messages was also applied to the remaining face-mask messages from the three data sources. As shown in Figure 3, most tweets (95%) and most Nu.nl comments (90%) are against the policy. There were too few Reddit posts on this topic to obtain accurate measurements, but the stance was also less than 30% supportive. A possible reason for the stance being more supportive on Nu.nl than on Twitter is that comments on Nu.nl are actively moderated while tweets are not, leaving less room for trolls to attack government policy. Another reason could be the general prevalence of polarized opinions on Twitter (DiResta, 2018). This finding also indicates the importance of capturing public reactions across different social media sources.

## 5 Discussion and future work

We present the preliminary results of sentiment analysis of Dutch social media data related to the COVID-19 measures across three sources. We concentrate on public polarity and stance towards two measures taken by the Dutch government against the spread of the corona virus. We found that a large number of messages on Dutch online media were related to the COVID-19 pandemic and the polarity of those messages tended to be more negative than general messages.

We assessed the stance of Dutch social media messages on the national advice on social distancing and wearing face masks. The analysis showed that people widely supported social distancing when the measure was announced in March, then support declined until June and increased again recently. We think this phenomenon may be related to the the pandemic situation in the Netherlands (the number of reported COVID-19 patients has decreased and increased during this time as well). With respect to face masks, analysis showed that the public think face masks are useful (against the governmental measure). The rejection rates remained relatively stable over the past months.

In future work we would like to improve the performance of our stance classification approach. As shown in Table 4, training set size restricts the performance of the classifier. Therefore we will start annotating more data using crowd-sourcing techniques. Furthermore, we aim to train a general cross-topic stance classifier to assess new topics. Transfer learning (Zarrella and Marsh, 2016) could be an interesting approach for this task.

From a practical perspective, we are interested in the explainability of our stance analysis results. We are collaborating with social scientists, who use quantitative methods (i.e. digital questionnaires and interviews) to investigate the influence of policy measures on the general public. A comparison study is planned to validate and explain the temporal trends of Dutch public sentiment.

## Acknowledgements

This work is funded by Netherlands eScience Center under the project PuReGoMe (27020S04).

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
