# OpenReview forum: "Public Sentiment on Governmental COVID-19 Measures in Dutch Social Media"
_EMNLP/2020/Workshop/NLP-COVID — NLP-COVID19-EMNLP Oral_

### Official Review · AnonReviewer3 · 2020-09-21
**Public Sentiment on Governmental COVID-19 Measures in Dutch Social Media**

**Rating:** 6
**Confidence:** 5

**Review:**

**SUMMARY**
This work described a series of public sentiment analysis regarding Dutch governmental policies. The study was based on the datasets from three types of social media: Twitter, Reddit, and Nu. Datasets are collected and filtered using a set of COVID related keywords from late February 2020 to July 2020. Sentiment analysis included two parts: 1) polarity analysis based on 3918 Dutch polarity words; 2) stance analysis--annotating and then classifying posts related to social distancing and face mask-wearing. For stance analysis, the trained classifier (with fastText) is applied to the entire dataset. The study showed the policy support value declined until June and then increased, which was consistent with the pandemic situation in the Netherlands.

**COMMENTS**
1. For the Reddit dataset in Table 2, is it the number of submissions(initiating posts)?
2. Reddit data is based on subreddits, can you please clarify the subreddits that you were using?
3. Is it trustworthy to have only one annotator without any adjudication process?

---

> ### Author Response · Authors · 2020-09-27
> **Reaction from the authors**
>
> Answers to questions:
>
> 1. Table 2 shows totals of the Reddit initial post and the comments
> 2. The subreddits used are coronanetherlands, CoronaNL and CoronavirusNL (all comments) and thenetherlands (only comments from the Megathread Coronavirus)
> 3. As the project is currently in progress, only annotations from a single annotator were available at the time of submitting the paper. Currently more annotators are involved in the labeling process. Analysis of agreement so far shows that the annotators agree strongly on the stance of a message against or in favor of government policy, which reinforces the results as presented in the paper. A source of disagreement is the judgement whether a message is relevant/contains a clearly expressed opinion - this annotation issue is currently being resolved as part of the ongoing research.

---

### Official Review · AnonReviewer1 · 2020-09-21
**Interesting approach and findings which could be improved by a more thorough analysis**

**Rating:** 6
**Confidence:** 4

**Review:**

This paper describes an analysis of COVID-19-related text messages in the Dutch language obtained from three online sources (Twitter, Reddit, Nu.nl; relevant messages were filtered using a selection of COVID-19 related keywords (Twitter) or based on specific topics (Reddit, Nu.nl)) from February to July 2020. Sentiment polarity analysis (using a lexicon-based approach) as well as stance detection for the topics of social distancing and the use of face coverings (using a supervised learning approach based on fastText) were conducted on the retrieved messages. For the latter, subsets of the retrieved texts were sampled based on keyword search and manually annotated by a single human annotator.
The findings show some alignments between COVID-19-related events and the corresponding reception on such social media platforms. For example, the first introduction of lockdown measures in the Netherlands in March 2020 led to a decrease in polarity on the Twitter data, and the first release measures in May 2020 caused the polarity to peak on the observed dataset.

The topic and approach presented in this paper can be very interesting to the community. However, it is unclear whether the collected online text data can be interpreted as representative for the “broader Dutch public” (as claimed in the paper) since only online data are used.

For the stance detection, the used dataset is relatively small (especially for the face covering topic), and it would be interesting to see how a simpler approach (e.g., a linear model using bag-of-words features) compares to the neural network-based approach.

Furthermore, some statistics (e.g., average sequence length) on the text messages obtained from the different sources would be helpful, since the stance detection system is trained only on parts of the collected data. In Figure 2, for example, the stance detection system is trained only on the Twitter data and evaluated on all three datasets.

It would also be interesting to see the polarity results on the other datasets (e.g., in the Appendix), only Twitter data are shown in the paper.

Questions:
1. Section 3.2: For the stance detection task, is there a reason for why only Twitter and Nu.nl data was used for the human annotation?
2. Which topics and articles were considered for the analysis of Reddit and Nu.nl data?

Comments:
1. In Section 2 different tenses are mixed up when discussing related work.
2. The axes in Figure 1 should be labelled and explained in more detail. Also, the letters A and B are hardly visible in the figure.
3. Section 3.2: Footnote should come after punctuation.
4. Section 4.1: Missing word “Some interesting links can [be] found…”

---

> ### Author Response · Authors · 2020-09-27
> **Reaction from the authors**
>
> Answers to questions:
>
> 1. Reddit data has only recently been included in the project, when the Twitter and Nu.nl annotations had already finished. Currently we have a second annotator working on annotating additional material from all three data sets
> 2. From Reddit we used the subreddits coronanetherlands, CoronaNL and CoronavirusNL (all comments) and thenetherlands (only comments from the Megathread Coronavirus). From Nu.nl we used the comments of all articles in the Coronavirus topic
>
> Response to the remark about the match between online data and the general public: Indeed messages on social media are not fully representative of the general public. However, opinions on social media do have a significant influence on the general debate in traditional media (tv, newspapers) and in society in general. Therefore we think that social media are suitable as a proxy for opinions of the broader public. Furthermore, the methodology using three very different social media contributes to the reduction of bias in opinions.

---

### Official Review · AnonReviewer2 · 2020-09-27
**Public Sentiment on Governmental COVID-19 Measures in Dutch Social Media**

**Rating:** 5
**Confidence:** 5

**Review:**

The paper describes an application of sentiment analysis for the dutch messages from three social platforms related to the COVID-19 measures.
The paper is well written, the introduced analysis is definitely needful and interesting. However, the paper has the following issues:
1.  Authors do not describe the classification algorithm that was used for the stance analysis. I realize that the authors applied an external tool, but still a brief explanation is needed.
2. According to my understanding, authors trained FastText word vectors on their data. However, authors do not explain why they do not use pre-trained vectors, despite the very well known fact that word embedding trained on a big data performs better than trained on a specific small data.  Also, the authors mix training word vectors with classification, which is confusing. If the used classification model contains an embedding layer, it also needs to be explained.
3. About manual annotation process.  Due to a noisy content in social media, sarcasm, and other factors that might affect the annotation, I assume that the annotator encountered some "difficult" cases. How these cases were handled?
4. What is the accuracy level of the used sentiment analysis (polarity)? Despite the method that was used is unsupervised, its evaluation on some test dataset is needed (or, if it was evaluated by developers on a similar domain, the relevant paper must be cited and its performance must be reported).
5. 10-fold cross validation is used for evaluation and NOT for training. Which model (trained on what?) was eventually used for automatic labeling of a new data?
6. Which classes were "difficult" for your classifier (were misclassified)? Accuracy per class can answer this question. Also, I think that reporting validation accuracy is redundant.

A small typo: "the COVID-19 related tweets *is->are* generally.. "

---

> ### Author Response · Authors · 2020-09-27
> **Reaction from the authors**
>
> Reaction to issues:
>
> 1. Good suggestion
> 2. We have evaluated using the pretrained vectors from Wikipedia provided with fastText. However, the performance gains obtained were too small in comparison to the increases in processing time so we have decided not to use them
> 3. The annotators have experienced few problems with deciding between positive and negative cases. The bulk of the problems of the annotators was related to deciding whether a tweet was relevant to the research question or not. It would indeed be interesting to discuss how the difficult cases were handled in a future longer paper
> 4. We have not been able to find relevant gold standard Dutch sentiment data so we have not evaluated the external sentiment tool. We could label sentiment data ourselves but we have chosen to focus our annotation efforts on stance annotation
> 5. The model applied for automatic model is a model trained on all available labeled data with the best parameters found during evaluation
> 6. We have not performed an error analysis yet but indeed that would also be interesting